# Unsupervised Approaches for the Segmentation of Dry ARMD Lesions in Eye Fundus cSLO Images

**DOI:** 10.3390/jimaging7080143

**Published:** 2021-08-11

**Authors:** Clément Royer, Jérémie Sublime, Florence Rossant, Michel Paques

**Affiliations:** 1ISEP—School of Digital Engineers, 92130 Issy-Les-Moulineaux, France; 2LIPN—CNRS UMR 7030, LaMSN—Université Sorbonne Paris Nord, 93210 St Denis, France; 3Clinical Imaging Center 1423, Quinze-Vingts Hospital, INSERM-DGOS Clinical Investigation Center, 75012 Paris, France; mpaques@15-20.fr

**Keywords:** dry ARMD, unsupervised learning, automatic segmentation, clustering, W-net

## Abstract

Age-related macular degeneration (ARMD), a major cause of sight impairment for elderly people, is still not well understood despite intensive research. Measuring the size of the lesions in the fundus is the main biomarker of the severity of the disease and as such is widely used in clinical trials yet only relies on manual segmentation. Artificial intelligence, in particular automatic image analysis based on neural networks, has a major role to play in better understanding the disease, by analyzing the intrinsic optical properties of dry ARMD lesions from patient images. In this paper, we propose a comparison of automatic segmentation methods (classical computer vision method, machine learning method and deep learning method) in an unsupervised context applied on cSLO IR images. Among the methods compared, we propose an adaptation of a fully convolutional network, called W-net, as an efficient method for the segmentation of ARMD lesions. Unlike supervised segmentation methods, our algorithm does not require annotated data which are very difficult to obtain in this application. Our method was tested on a dataset of 328 images and has shown to reach higher quality results than other compared unsupervised methods with a F1 score of 0.87, while having a more stable model, even though in some specific cases, texture/edges-based methods can produce relevant results.

## 1. Introduction

Age-related macular degeneration (ARMD) is a degenerative disease that affects the retina, and a leading cause of visual loss.

In this paper, we focus on the dry form of this pathology which currently does not have any treatments. It is characterized by a progressive loss of pigment epithelium which engenders a lesion located in the macula, growing slowly and impeding more and more the patient central vision. The lesions, called geographic atrophy (GA), can be observed in eye fundus images. Figure 1 shows examples of confocal Scanning Laser Ophtalmoscopy (cSLO) images acquired in infrared (IR), a commonly used imaging technique for ARMD patients, where the GA appears as brighter areas.

Despite intensive biological research, the factors involved in progression are poorly known. Therefore, clinical studies are needed to characterize the disease and its evolution. This can be done from eye fundus images, which are routinely acquired during the medical follow-up of patients. However, there are at present no efficient algorithms to automatically process large databases of images, even though it is very costly to process them manually: first, the manual delineation of the lesions is a very difficult and time-consuming task, given the complex structure of the GA. Secondly, the reliability of manual delineations is also an issue as even experts tends to disagree on their segmentations [1]. To solve this problem, in this work, we propose the first fully unsupervised application of automatic segmentation of GA using W-net [2] on cSLO IR acquired images (Section 2) and to assess how well it performs compared with other state of the art unsupervised methods. Our contribution is therefore three-fold:First, we propose a successful adaptation of the original developed by Xia et al. [2]. We modified the architecture to adapt it to our images and their specifics. In addition, furthermore, we fully trained our network and did not use any pre-trained model.Second, we propose the first realistic unsupervised approach to the very difficult problem of ARMD lesion segmentation. Indeed, this problem is already difficult for humans, and has very little labelled data (hence why we cannot use supervised neural networks), thus making it quite a challenging problem for unsupervised algorithms. In this regard, we achieve very decent performance considering the nature of the problem and the challenges it presents.Third, we propose a fair and extensive comparison with other unsupervised methods (neural networks and others) used in other fields that we have also adapted to tackle the same problem.

The remainder of this paper is organized as follows: Our method is presented in Section 4.1 and the compared methods in Section 4.2, after the description of our dataset (Section 2) and the related works (Section 3). The experimental results are shown in Section 5. In addition, finally, conclusions and insights as to our future works are discussed in Section 6.

## 2. Materials

We work on cSLO images acquired in IR. This modality is more robust and less invasive than fundus autofluorescence (FAF), and it has higher resolution and higher contrast between normal and diseased areas than color imaging, an older technology. Our database is composed of 18 series of images (328 images in total), showing the progression of the GA over several years for 13 patients (ARMD can affect both eyes). The time interval between two exams is about 6 months. All these images has been obtained from the Clinical Imaging Center of the Quinze-Vingts Hospital and all patients gave their informed consent for the use of images for clinical studies.

All images are in gray-levels (1 channel). Black borders are present because of the spatial registration process. Moreover, the images present illumination artifacts, blur and a non-uniform illumination. We preprocess jointly the images of every series as follows: we crop the images to suppress the black borders and resize them to 256 × 256 pixels; then we apply a joint correction of the overall illumination [3] so that any two processed images have comparable gray-levels. This algorithm does not completely compensate for uneven illumination, but the remaining defect is the same in all processed images, helping the ophthalmologists in the visual comparison of any two exams. From the segmentation point of view, the algorithm homogenizes the luminosity and the contrast of the images applied to the W-net and thus reduces the radiometric variability.

The major challenge of the segmentation task is to adapt to the large and complex variability of GA structures and to cope with all kinds of image defects. Figure 1 illustrates the difficulty: variability in size, shape, texture, number of GA areas, with new lesions that may appear at any time or merge; low contrast, blur, high ambiguity in the actual position of the border even for the most contrasted images. Please note that 18 series of images may not be enough to fully represent the real variability of GAs, and can lead to a lack of generality for deep learning-based methods.

In order to evaluate our algorithm, we asked ophthalmologists to delineate the GA areas. However, even with expert skills, the produced annotations may not be 100% reliable. Thus, the resulting pixel-wise annotations are only used to calculate classic segmentation quality measures (see Section 5.1).

## 3. Related Work

A lot of research has been done to propose segmentation algorithms of the GA. Standard algorithms have been applied, such as region growing [4,5] or active contour [6], region oriented variational methods with level set implementation [7,8], texture analysis [9] and watershed segmentation [10,11]. Most of the proposed methods are based on supervised machine learning approaches, with statistical models [12], random forests [13], random forests combined with Support Vector Machine [14] or k-nearest neighbor classifiers [15]. However, even in a supervised context, it is intricate to obtain a fully automatic algorithm reaching the required level of performance and some authors add human interaction to guide their algorithm [16,17]

Using deep learning, a reference model achieving impressive results for supervised segmentation is the U-net [18], and it has been implemented in many medical image segmentation problems (e.g., [19,20]). The U-net takes advantage of residual connections combined with a contracting and an expansive part. The authors of [21] proposed a supervised algorithm to follow-up the GA progression, using U-nets to first segment vessels and the optic disc, reducing the region of interest for the GA detection and then tracking it using intensity ratio between neighbor pixels. Other supervised deep learning-based methods were applied on ARMD as in [22] where they exploit transfer learning with deep neural networks to detect ARMD.

Last but not least, a segmentation task has been investigated in a scene parsing context. The authors of [23,24] exploit spatial pyramid pooling architecture in order to perform semantic segmentation between multiples objects present in a scene. However, as the U-Net, both PSP-Net and APSP-Net are deep supervised methods and their training requires a large amount of annotated data, which is not suitable for our application, as mentioned previously.

Thus, unsupervised methods can solve both the problem of data availability and the issue of the reliability of experts’ annotations. Unsupervised automatic segmentation is mainly handled in two steps: extracting features and then applying a clustering algorithm. The authors of [25] applied fuzzy CMeans clustering, which reached good performance for high-contrast FAF images, but performed less well for other modalities.

Using deep learning, the authors of [26] applied a joint auto-encoder (initially applied on satellite images [27]) on the same dataset we are using in this paper (see Section 2), in order to perform automatic change detection in an unsupervised context learning. The algorithm outperformed the state-of-the-art; however, as this model aims to detect changes, it is not comparable with automatic segmentation algorithm.

Kanezaki [28] combined clustering algorithm and CNN for a fully unsupervised model using a superpixel refinement method and achieving promising results for unsupervised automatic semantic segmentation. For a given image, the CNN first oversegments it (high number of cluster initially), then at each iteration, tries to reduce the number of cluster by merging them according to the result of the clustering algorithm.

Table 1 summarizes the pros and cons for the related works. As we want to exploit the dataset provided by the Clinical Imaging Center of the Quinze-Vingts Hospital (Section 2), which represent a non-invasive modality acquisition adapted to the patient follow-up, other modality use (which can engender better results due to a better quality of imaging) will be considered as a drawback. Supervised training is also considered a weakness because of the annotations reliability.

In this paper, we focus on unsupervised algorithms to segment the GA in IR cSLO eye fundus images with dry-ARMD. We propose to adapt the W-net [2] as well as three other unsupervised methods and compare them in an experimental study.

## 4. Methods

### 4.1. Our Method: W-Nets Adapted to ARMD Lesions Segmentation

The W-net model is a fully convolutional autoencoder for which both encoder and decoder are U-net networks [18]. While U-nets are supervised neural networks commonly used for image segmentation, W-nets are the unsupervised equivalent. Unlike U-Nets, W-nets are trained using two loss functions. The first one is the usual reconstruction error used to train classic autoencoder, the second one is the soft-N-cut loss [2], a smooth version of the N-cut loss [29]. Minimizing the soft N-cut loss has the effect of enhancing the segmentation quality by maximizing the dissimilarity between the different clusters. The dissimilarity calculated is a function of the intensity pixel and the spatial position of the pixels.

During each training step, we have two successive optimization steps after each forward. First the full W-net is updated by back-propagating the self-reconstruction error ($MSE=\frac{1}{n}\sum_{k=1}^{n}{\left(Y_{k}-{\hat{Y}}_{k}\right)}^{2}$, where *n* is the number of pixel in the image), then only the encoding part is updated based on the soft N-cut loss :$$\begin{array}{cc} J_{soft-Ncut}\left(V,K\right) & =K-\sum_{k=1}^{K}\frac{\sum_{u\in V,v\in V}w\left(u,v\right)p\left(u=A_{k}\right)p\left(v=A_{k}\right)}{\sum_{u\in A_{k},t\in V}w\left(u,t\right)p\left(u=A_{k}\right)} \end{array}$$
with $p(u=A_{k})$ the probability for the pixel *u* to belong to the class *k*, and *w* a function which compute a weight for each couple of pixel, based on their position in the image and their intensity [29].

The architecture of our W-net is presented in Figure 2. Hence, the entire training of the W-net is unsupervised: the MSE loss is computed with the W-net output and the image input, while the soft-N-cut loss is computed with the probability map produced by the encoder (after the softmax layer in Figure 2) and the weights (which only need the image input to be computed). Compared with the original one, an average-pooling layer has been added before the computation of the soft N-cut loss. This loss is based on graph calculation, and thus, is memory-consuming when using large size images and not applicable on 256 × 256 sized images. Hence, the value of the soft N-cut loss is approximated by the value of the loss applied on the 128 × 128 (average) pooled input.

Once it has been trained on a set of unlabeled images, our network can be used on any ARMD cSLO images acquired in IR that have a similar resolution: For each pixel, the network will provide the probability of belonging to each class. Then applying a simple argmax will produce a segmentation. In our case, the purpose is to obtain two classes corresponding to the GA and the retina background. Thus, when the W-net is set with K=2, one segmentation class will represent the GA and the other one the retina background. However, when the W-net is set with K>2, we have to map the extra classes to the GA class or the background class. Finally, once trained, our W-net will always produce the same semantic classes for every image it processes, thus requiring a manual mapping only once.

### 4.2. Compared Methods

In this subsection, we present the three other compared algorithm: Gabor filters with KMeans [30], an active contour based model [5] and a model that combine CNN and clustering [28].

#### 4.2.1. Gabor + KMeans

This algorithm is divided in three steps: creation of a set of Gabor filters, feature extraction using previous filters, pixels classification with KMeans on those features.

For a given frequency $u_{0}$, Gabor function in spatial domain is a Gaussian modulated with a sinusoïd:$$h\left(x,y\right)=\frac{1}{2\pi \sigma_{x}\sigma_{y}}exp\left\{-\frac{1}{2}\left[\frac{x^{2}}{\sigma_{x}^{2}}+\frac{y^{2}}{\sigma_{y}^{2}}\right]\right\}.cos\left(2\pi u_{0}x\right)$$
with $\sigma_{x}$ and $\sigma_{y}$ variances in direction *x* and *y*.

To capture features in multiple direction, we can add a rotation θ to the filter. Thus, we chose a set of directions, which define the set of Gabor filters. Each filter output pass through a non-linear function (sigmoïd function). The texture features are completed with spatial information (pixel position).

Finally, a KMeans clustering algorithm is applied on all the features, classifying each pixel and thus given us a pixel-wise segmentation.

#### 4.2.2. Active Contour Model without Edges

Chan and Vese combined in [5] active contour methods, and Mumford-Shah functional and level set methods. Parametric active contour methods are based on a curve evolving technique: a parametrized curve $C:\left[0,1\right]⟶R^{2}$ evolve and is stopped using an edge detector, which usually rely on the gradient |∇I| for a given image *I*. Such a method can then only detect objects with high gradient on their edges. Thus, the authors of [5] proposed a region oriented approach based on Mumford-Shah energy functional.

Let *C* be the evolving curve, *I* a given image, $c_{1}$ and $c_{2}$ the average value of *I* respectively inside *C* and outside *C* (case example with one object to detect from the background). The fitting energy function is defined as:$$F_{1}\left(C\right)+F_{2}\left(C\right)=\int_{inside(C)}|I\left(x,y\right)-c_{1}{|}^{2}dxdy+\int_{outside(C)}{\left|I\left(x,y\right)-c_{2}\right|}^{2}dxdy$$

Thus, the solution of $\inf_{C}\left\{F_{1}\left(C\right)+F_{2}\left(C\right)\right\}$ gives the boundary of the object we want to detect.

This algorithm is suitable for the GA segmentation as it can handle changes in topology with respect to the initialization, thanks to the level-set resolution method, as well as low contrast along the edges.

#### 4.2.3. CNN + Superpixel Refinement

Kanezaki investigate the use of CNN for unsupervised segmentation. The algorithm is based on three assumptions:“Pixels of similar features are desired to be assigned the same label”“Spatially continuous pixels are desired to be assigned the same label”“The number of unique cluster labels is desired to be large”

For a given pixel $v_{n}$ of an RGB image (three channels) $I={\left\{v_{n}\in R^{3}\right\}}_{n=1}^{N}$ that we want to segment into *q* classes, we first extract *p*-dimensional features $x_{n}$ with *p* filters of size 3×3. Applying a linear classifier $f_{c}$ with weights $W_{c}\in R^{p\times q}$ and bias $b_{c}\in R^{q}$ gives us a response map for each pixel: ${\left\{y_{n}=W_{c}x_{n}+b_{c}\right\}}_{n=1}^{N}$. Finally the cluster $c_{n}$ for each pixel is obtained by selecting the dimension with the greatest value. Thus, the output model is consistent with the features similarity constraint.

To satisfy the spatial continuity constraint, *K* fine superpixels ${\left\{S_{k}\right\}}_{k=1}^{K}$ are extracted and all the pixels of a given superpixel is forced to have the same cluster (which will be the most frequent cluster in the superpixel). Slic algorithm (KMeans-based segmentation algorithm) is used to compute the superpixels. Hence, the *K* classes obtained with Slic algorithm applied on the image *I* will correspond to the *K* superpixels.

The method in [28] aim to segment natural images (experiments on Berkeley Segmentation Dataset and Benchmark BSDS500) therefore, we do not have any prior knowledge of the number of unique clusters present in a given image. In this context, the algorithm must be able to output a variable number of clusters. Kanezaki’s strategy is to constraint the number $q^{\prime }$ of output clusters with a maximum number of cluster *q*. One has $1\leq q^{\prime }\leq q$ and to avoid the naive solution $q^{\prime }=1$, he introduced the last constraint: *q* is preferred to be large. This is done by adding a whitening process, which transforms the response map $y_{n}$ into $y_{n^{{}^{\prime }}}$ where each axis has zero mean and unit variance.

Two steps are applied alternatively to train the network:forward process: prediction of clusters $c_{n}$ with the network and refined cluster $c_{n}^{\prime }$ with the superpixel refinement processbackward process: backpropagation of the signal error (cross-entropy loss) between the network response $y_{n}^{{}^{\prime }}$ and the refined cluster $c_{n}^{\prime }$

The training process of this model is different from the W-net one. One has to initialize and train the model for each individual image at contrary to the W-net which has to be trained only once. Indeed, the training process corresponds to the refinement of the model on a given image and can not be used to infer an other image.

## 5. Results

### 5.1. Experimental Setting

In our experiments, we use 18 series of 5 to 52 images (328 images). Due to the relatively small number of series and their variable size, we could not use k-folds validation. However, we used 8 different random combinations of 12 series to train the model and 6 for the tests. Our W-net is compared with Gabor Filters for texture extraction [30] and KMeans algorithm for the segmentation, the CNN model with superpixel refinement [28] and finally an active contour model [5].

As the lesions can only grow bigger, we also try to see if adding the segmentation from the previous image improves the results or not: we merge the segmentation provided by our W-net for the current Image $I_{t}$ with the segmentation of the same W-net for the previous image $I_{t-1}$. This modification is supposed to reduce the risk of undersegmentation.

Our network is set with a kernel size of 3, and each U-net produces 1024 features after the contracting part. Our model is trained on 250 epochs, using two Adam optimizers. The number of classes is set to 3 in the experiments: using two classes, the latent representation is too restrictive. Using three classes, the network learns extra classes for the background (healthy regions and blood vessels) or the ARMD lesions.

Hyperparameters for Gabor filters methods, Chan and Vese’s method, and Kanezaki’s method have been tuned manually by referring to the visual output. Therefore, the fixed parameters are not the optimal ones, but a compromise to obtain the best average output.

The evaluation is based on the pixels’ classification using true positive (TP), false positive (FP), and false negative (FN) to build dice metrics such as the F1-Score:$$F1=2\times \frac{precision\times recall}{precision+recall}=\frac{TP}{TP+0.5(FP+FN)}$$
with
$$precision=\frac{TP}{TP+FP}$$
and
$$recall=\frac{TP}{TP+FN}$$

KMeans algorithm output directly the classification an not the probability to belong to each class. However, those probabilities are needed to compute the ROC curve. Hence we use fuzzy CMeans instead (which provides, with an argmax, the same output as Kmeans)

All the experiments have been done using a GEFORCE RTX 3090 GPU with 24Gb of RAM, Python 3.8.5, Pytorch 1.7.1 and Cuda 11.1.

### 5.2. Experimental Results

As explained before, our W-net is set to three classes. With this configuration, we manually map the classes to the lesion or the background. Figure 3 shows an example of different classes obtained with our 3-class W-net: the class of interest corresponding to the GA is in green. The two other classes belong to the retina background. Please note that this mapping is arbitrary and may not be optimal in specific cases.

The results are shown in Table 2 and Table 3.

Table 2 shows the average dice score (F1), precision and recall on the training set (W-net is the only model that need to be trained), while Table 3 focuses on the test set and features the comparison with the other algorithms introduced in Section 5.1. As one can see, W-net is the most relevant method. It has the best F1 score and achieves higher quality results with a better precision and recall. Moreover, our W-net has a smaller standard deviation for the F1 score and the precision, resulting in a more stable model.

The bottom line (W-net + $Seg_{t-1}$) in both tables corresponds to the fusion of a given segmentation with the previous one (see Section 5.1). However, we can see that this modification does not lead to any significant increase in the result quality: while the Recall is indeed better (less under-segmentation), the F1 score is not significantly different from the one of our method on individual images. This can be explained by an accumulation of over-segmented areas propagating through the series, which is confirmed by a worst Precision that we observe.

In any case, we can see that except for the recall—which can be explained by cases of under-segmentation—our W-net approaches outperform the Gabor filter and KMeans approach, Active contour method, and CNN model with superpixel refinement.

Our W-net superiority is also shown in Figure 4, where we can clearly see that the W-net is nearer to the perfect model compared to the Gabor filter with Fuzzy CMeans method. Please note that we compared the ROC curve for the two most relevant algorithm according to the results in Table 2 and Table 3.

Table 4 shows detailed score on 6 series and we can see an example taken from each series in Figure 5, Figure 6, Figure 7, Figure 8, Figure 9 and Figure 10.

Figure 5 and Figure 9 show examples from average cases (in terms of complexity of the lesion’s structures), where W-net outperforms other methods with a mean F1 score on the series of 0.98 for the first series, and 0.946 for the second one. This also highlights the capacity of W-net to correlate the different classes based on intensity values, while the other methods attempt (and fails) to do so based on textures rather than intensity.

As said in the introduction, a lesion with blurred outlines will be difficult to segment. This is illustrated in Figure 6 where the Gabor filter algorithm fails to segment the GA, while the W-net does better but with some undersegmentation. In this case, CNN with super-pixel refinement method and the active contour methods output a better result; however, the first one gives more classes than expected due to the algorithm’s setting [28].

Finally, Figure 7 and Figure 10 show two cases where the lesion has different levels of contrast. These are good examples of cases in which it is useful to have three classes (which are represented in yellow/green/purple for example d): as one can see, most of the lesion class is contained within the green class, which is one of the two retina background classes, while the yellow class is the GA class (according to our mapping). In those cases and without the manual cluster/class mapping, both Gabor method and our W-net would fail to identify the GA. The W-net reliance on intensity creates this undersegmentation risk: a low-intensity region is more likely to belong to the background retina class. However, because of the extra class, in the case of our method, the segmentation can still easily be manually fixed. In this case, Kanezaki’s method and Chan and Vese’s method provide a better quality results. Nevertheless, it highlights the limits of unsupervised methods where user intervention is sometimes inevitable.

We can also mention that W-net outperfoms other compared algorithm on a long serie (Patient Id 020, fourth line of Table 4, Figure 9) which contain 50 images. This serie is an example of ARMD progression: as we can see in Figure 1e, the ARMD lesions progress from two tiny GA to three consequent GA in 6 years. Images from the beginning of the serie thus contain small GA which enhances risk of oversegmentation. This can also be seen in Figure 8 where despite a higher quality result provided by our W-net, it did not detect a small GA and oversegmented the top right region of the image.

From these experiments, we can see that in most cases, our modified W-net produces a better quality segmentation than the other algorithms and has the advantage to be more stable. Furthermore, from cases such as Figure 6, we can see that because of its reliance on textures only, the Gabor filter based approach sometimes entirely fails to capture the GA and appears to detect only peripheral areas of textural transition. However, our proposed W-net -despite some mismatches between the clusters and the classes- manages to have a group of clusters that match the GA class. The CNN with superpixel refinement method provides relevant outputs but need a post-processing step to enhance the segmentation quality. In fact, the results are not smooth enough, and we have no guarantee to obtain two final clusters: the number of clusters in the outuput can vary from an image to another (even for two successive images from a given serie). This is due to the third criterion of Kanezaki’s system: the segmentation context is different from ours, we aim to segment a fixed number of classes. Last but not least, active contour methods can produce a better quality result in some specific cases as they can totally fail in an average case. That highlight the lack of generality of those methods compared to the high variability of both texture and structure of the GA regions.

## 6. Conclusions and Future Works

The automatic segmentation of dry-ARMD lesion is a difficult problem with significant implications for patients afflicted by this disease, as it may allow for a better monitoring of the lesions progression. Despite the complexity of the task, we proposed a promising adaptation of W-nets to this problem. Our algorithm is fully unsupervised, which makes it possible to achieve good quality results without any need for labeled data that are difficult to come by, as even expert ophthalmologists often disagree on the proper segmentation of the lesions for difficult cases.

Our proposed algorithm was tested on 328 images from 18 patients, and has proven to be very effective. However, we identified a weakness, which is the W-net’s tendency to undersegment the lesions. This can be explained by the intensity distribution in the images: a single-channel input model can only exploit pixel intensity and spatial information. Thus, high intensity regions will generally correspond to be a part of a lesion, while low intensity regions generally correspond to be a part of the background. However, there is a high variability in the contrast in the images and some cases can show a GA with an intensity barely lighter than the background, hence the interest of having three classes. Furthermore, the most difficult case for this task is when the lesion has a blurred outline (e.g., Figure 1(a.1)). Automatic unsupervised segmentation algorithms tend to fail in this specific case in two different ways: if the blurry lesion is too small, the result will be over-segmented. If the blurry lesion is too large, the result will be under-segmented.

On the other side, the three other algorithms have a lack of generality because of the restrictive settings. To better exploit their advantage, a proper fine tuning in each specific case should be needed, leading to a supervised process that requires medical expertise.

Because of the reliability of the ground truths and the cost to produce them, the unsupervised context is the most appropriate to face this issue

In our future work, we plan on improving our segmentation capabilities by combining our proposed algorithm with generative adversarial networks [31] that have been shown to produce higher quality outputs and to outperform traditional convolutional autoencoders. Thus, a first improvement of the W-net is an adversial training, which could improve the reconstruction quality and therefore the segmentation map quality.

Finally, deep learning performance depends a lot on the dataset training, thus enrich the dataset with more series and with data augmentation can enhance robustness and performance.

## Figures and Tables

**Figure 1 jimaging-07-00143-f001:**
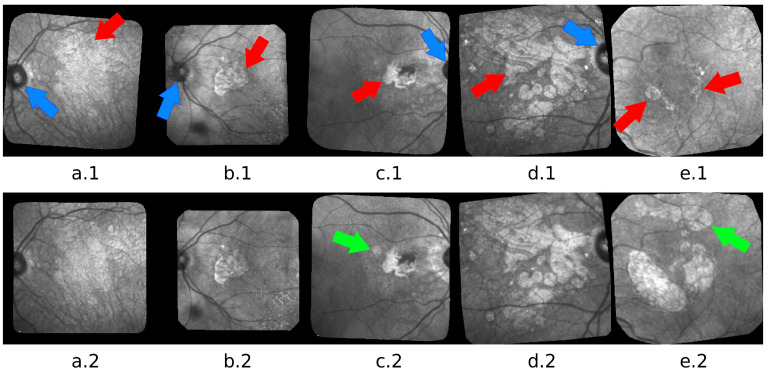
Five pairs of images: (**a**) advanced case with blur, low contrast and very indented GA boundary; (**b**) less advanced case with lesions at the center and around the optic disk; (**c**) a new lesion appears in the second image (green arrow); (**d**) GA with complex structure and texture; (**e**) example of progression of a GA during 6 years. A third GA appeared (green arrow). Arrows point to the GA (red) and to the optic disk (blue).

**Figure 2 jimaging-07-00143-f002:**
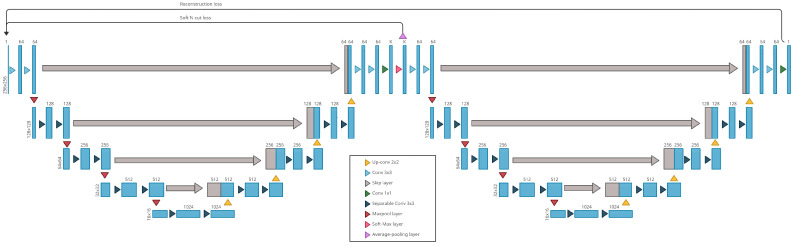
Our W-net architecture for ARMD lesions segmentation.

**Figure 3 jimaging-07-00143-f003:**
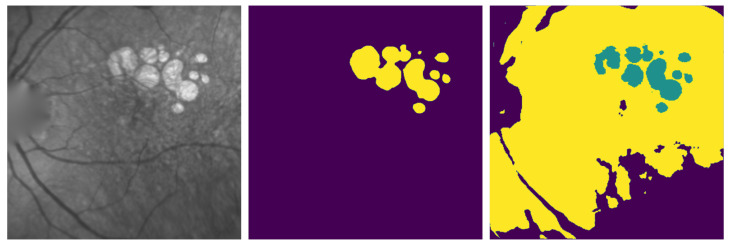
Example of three classes W-net output. From left to right: original image, ground truth, 3-class W-net output.

**Figure 4 jimaging-07-00143-f004:**
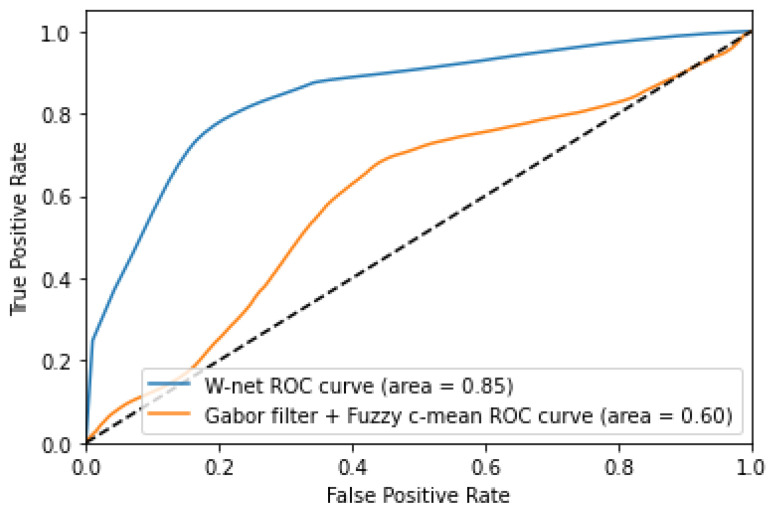
Receiver Operating Characteristic curve of Gabor filter + fuzzy CMeans and W-net.

**Figure 5 jimaging-07-00143-f005:**
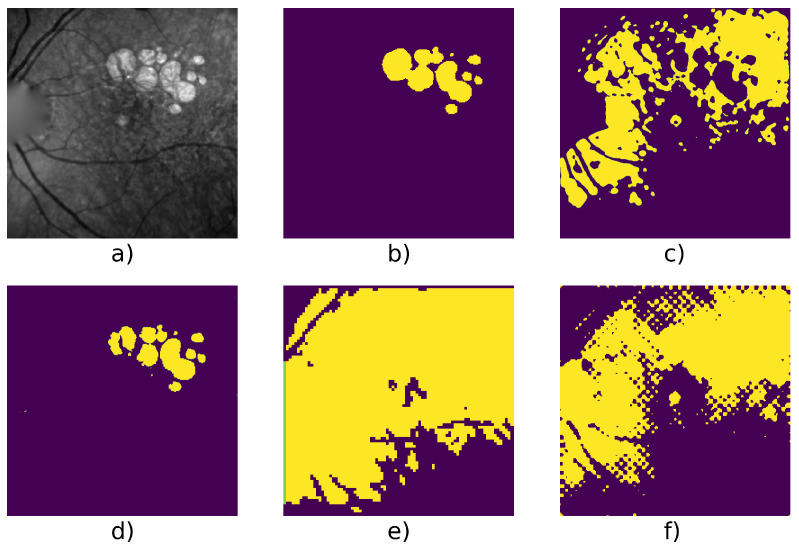
Patient id 117: (**a**) original image, (**b**) ground truth, (**c**) Gabor method segmentation, (**d**) W-net segmentation, (**e**) CNN + superpixel refinement segmentation, (**f**) active contour segmentation.

**Figure 6 jimaging-07-00143-f006:**
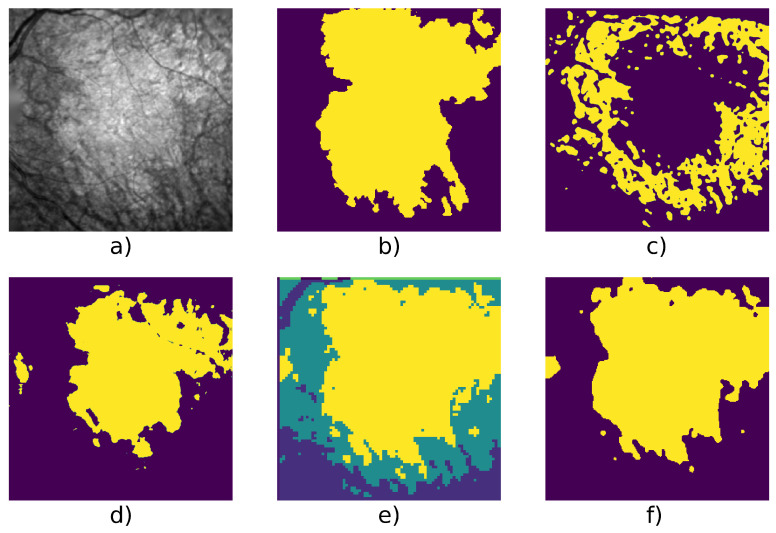
Patient id 005: (**a**) original image, (**b**) ground truth, (**c**) Gabor method segmentation, (**d**) W-net segmentation, (**e**) CNN + superpixel refinement segmentation, (**f**) active contour segmentation.

**Figure 7 jimaging-07-00143-f007:**
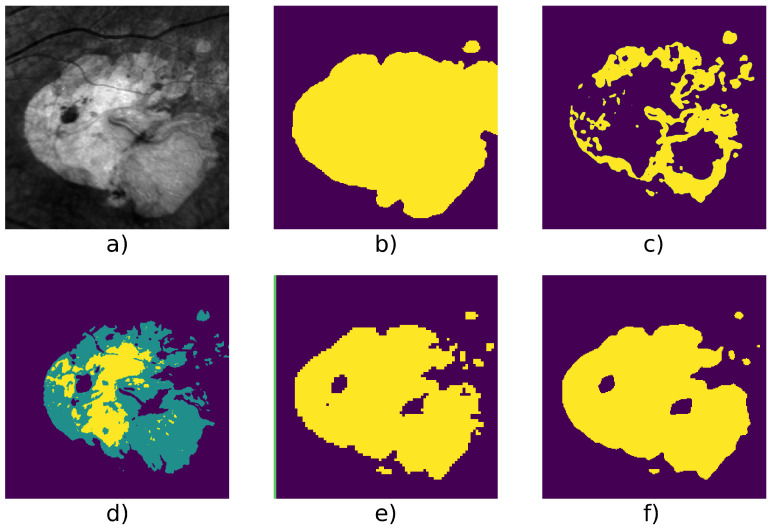
Patient id 016: (**a**) original image, (**b**) ground truth, (**c**) Gabor method segmentation, (**d**) W-net clusters, (**e**) CNN + superpixel refinement segmentation, (**f**) active contour segmentation.

**Figure 8 jimaging-07-00143-f008:**
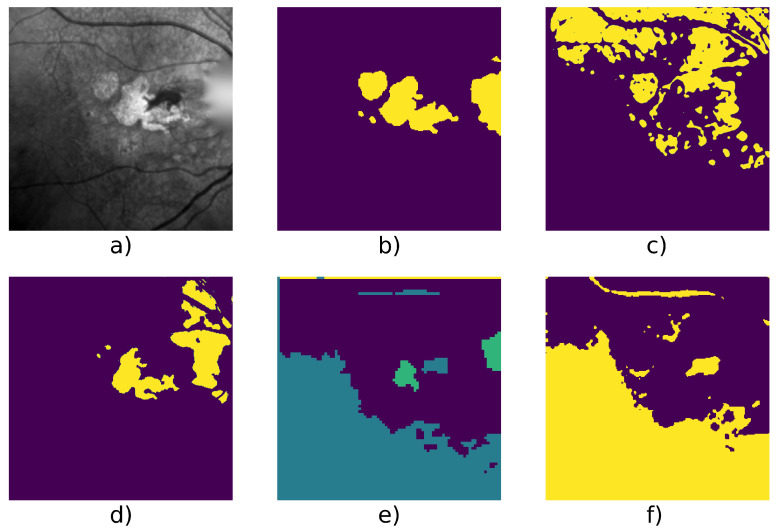
Patient id 010: (**a**) original image, (**b**) ground truth, (**c**) Gabor method segmentation, (**d**) W-net segmentation, (**e**) CNN + superpixel refinement segmentation, (**f**) active contour segmentation.

**Figure 9 jimaging-07-00143-f009:**
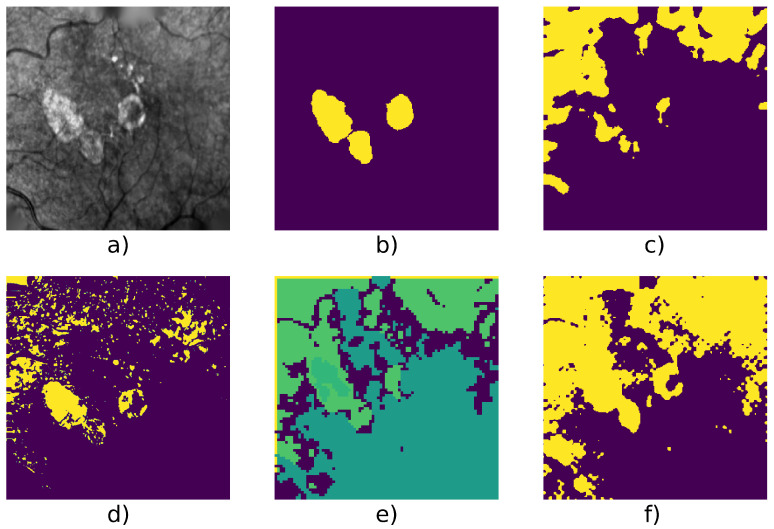
Patient id 020: (**a**) original image, (**b**) ground truth, (**c**) Gabor method segmentation, (**d**) W-net segmentation, (**e**) CNN + superpixel refinement segmentation, (**f**) active contour segmentation.

**Figure 10 jimaging-07-00143-f010:**
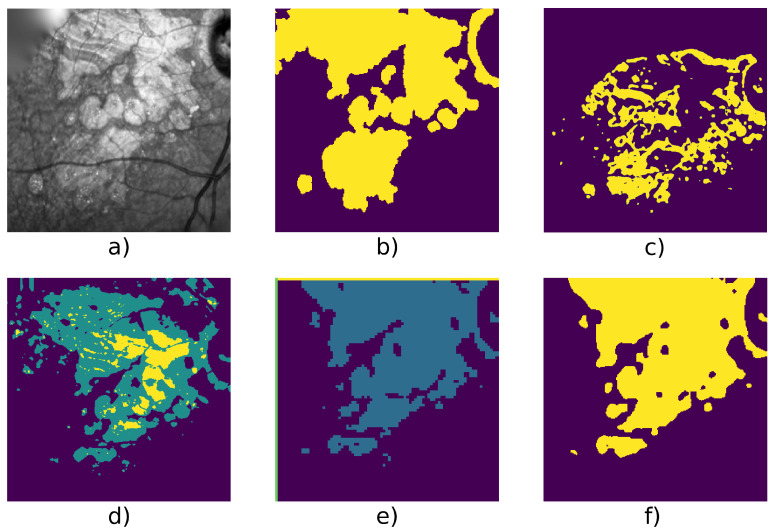
Patient id 109: (**a**) original image, (**b**) ground truth, (**c**) Gabor method segmentation, (**d**) W-net clusters, (**e**) CNN + superpixel refinement segmentation, (**f**) active contour segmentation.

**Table 1 jimaging-07-00143-t001:** Comparative summary of related works.

Method	Pros	Cons	
Region oriented [4,5]	High performance on ARMD, semi-supervised (seeds)	FAF/OCT images	Conventionnal methods
Active contour [5,6]	High performance on retinal cases	Segment optic discs	
Statistical [12]	High performance on retinal cases	Segment blood vessels and optic discs	
Random Forest [13]	High performance on ARMD	Color fundus images, supervised	
Random Forest + SVM [14]	High performance on ARMD	Screening and grading task, supervised	
Fuzzy C-means [25]	Unsupervised, high performance on ARMD	High contrast FAF images	
K-NN [15]	High performance on ARMD	FAF images, supervised	
Watershed [10,11]	Semi supervised (seeds)	OCT images	
U-net [18,21]	High performance on ARMD	Supervised, training on GPU	Deep learning methods
Transfert learning on ARMD [22]	High performance	Supervised, color fundus images	
Scene parsing [23,24]	High performance	Supervised, requires multiple objects in a scene, training on GPU	
Change detection [26]	Unsupervised, applied on the same dataset	Change detection task	
CNN + Superpixel refinement [28]	Unsupervised, no training	Produce a variable number of cluster in the segmentation	
W-net [2]	Unsupervised, robust	Training on GPU	
Our W-net	Unsupervised, fast inference use, robust, high performance on ARMD	Training on GPU	
Human interaction [16,17]	High performance on ARMD	Require human interaction	Other methods

**Table 2 jimaging-07-00143-t002:** Average W-net dice scores on the training set.

Method	F1	Precision	Recall
W-net	**0.83 ± 0.09**	**0.87 ± 0.08**	0.81 ± 0.13
W-net + $Seg_{t-1}$	0.82 ± 0.07	0.82 ± 0.10	**0.82 ± 0.11**

**Table 3 jimaging-07-00143-t003:** Average dice scores on the test set.

Method	F1	Precision	Recall
Active contour (Chan & Vese [5])	0.73 ± 0.07	0.64 ± 0.13	0.86 ± 0.05
CNN + Superpixel Refinement (Kanezaki [28])	0.65 ± 0.07	0.54 ± 0.10	0.85 ± 0.06
Gabor + KMeans [30]	0.77 ± 0.08	0.80 ± 0.12	0.75 ± 0.08
Our W-net	**0.87 ± 0.07**	**0.90 ± 0.07**	0.85 ± 0.11
W-net + $Seg_{t-1}$	0.85 ± 0.06	0.84 ± 0.07	**0.87 ± 0.09**

**Table 4 jimaging-07-00143-t004:** Detailed dice scores on specific series.

Patient Id	Method	F1	Precision	Recall	Nb. of Images	Fig.
005	Active Contour	0.787	0.779	0.795	9	Figure 6
	CNN + Superpixel refinement	0.690	0.623	0.787		
	Gabor + KMeans	0.791	0.760	**0.828**		
	Our W-net	0.785	**0.806**	0.765		
	W-net + $Seg_{t-1}$	**0.799**	0.805	0.792		
010	Active Contour	0.644	0.504	0.892	6	Figure 8
	CNN + Superpixel refinement	0.589	0.440	0.909		
	Gabor + KMeans	0.809	0.907	0.731		
	Our W-net	**0.922**	**0.921**	0.922		
	W-net + $Seg_{t-1}$	0.919	0.910	**0.927**		
016	Active Contour	**0.849**	**0.869**	0.828	31	Figure 7
	CNN + Superpixel refinement	0.790	0.752	0.840		
	Gabor + KMeans	0.678	0.596	0.786		
	Our W-net	0.676	0.880	**0.924**		
	W-net + $Seg_{t-1}$	0.706	0.817	0.622		
020	Active Contour	0.654	0.516	0.901	50	Figure 9
	CNN + Superpixel refinement	0.622	0.489	0.898		
	Gabor + KMeans	0.744	0.903	0.640		
	Our W-net	**0.946**	**0.977**	0.920		
	W-net + $Seg_{t-1}$	0.864	0.808	**0.929**		
109	Active Contour	0.774	0.755	**0.796**	16	Figure 10
	CNN + Superpixel refinement	0.717	0.695	0.767		
	Gabor + KMeans	0.700	0.685	0.718		
	Our W-net	0.782	**0.936**	0.672		
	W-net + $Seg_{t-1}$	**0.799**	0.903	0.717		
117	Active Contour	0.658	0.512	0.920	6	Figure 5
	CNN + Superpixel refinement	0.609	0.469	0.892		
	Gabor + KMeans	0.933	0.966	0.902		
	Our W-net	0.987	**0.995**	0.979		
	W-net + $Seg_{t-1}$	**0.988**	0.993	**0.982**		

## Data Availability

Sample images and the source code are available in https://github.com/clement-royer/Unsupervised-segmentation-of-dry-ARMD-lesions-in-cSLO-images-using-W-nets (accessed on 25 June 2021). Please note that since we are dealing with medical images, we are only allowed to disclose a limited number of them and not the full time series.

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
