# Peer review of "Unsupervised Approaches for the Segmentation of Dry ARMD Lesions in Eye Fundus cSLO Images"

_2313-433X, 2021, doi:10.3390/jimaging7080143_

Round 1
Reviewer 1 Report
The Authors presented unsupervised approaches for the segmentation of dry age-related macular degeneration lesions in eye fundus cSLO images. The topic is interesting. Even though the article is interesting in its current format, some aspects should be improved for possible publication and for a better understanding by the readers. Comments formulated during my review are presented below. These are as follows:
1) As the image segmentation is the significant technique in this paper, the authors should briefly provide several groups of image segmentation methods used in medical imaging, namely:
*superpixel segmentation methods [a,b]
*watershed segmentation methods [c,d]
*active contour methods[e,f]
[a] "Superpixels: An evaluation of the state-of-the-art". Comput. Vis. Image Underst. 166, (2018): 1-27.
[b] "Automated layer segmentation of macular OCT images via graph-based SLIC superpixels and manifold ranking approach". Computerized Medical Imaging and Graphics, 55, (2017): 42-53.
[c] "Watershed cuts: thinnings, shortest path forests, and topological watersheds", IEEE Trans. Pattern Anal.Mach. Intell. 32, (2010): 925-939.
[d] "Marker controlled watershed transform for intra-retinal cysts segmentation from optical coherence tomography B-scans". Pattern Recognition Letters, 139, (2020):86-94
[e] "An edge-based active contour model using an inflation/deflation force with a damping coefficient". Expert Systems with Applications, 44, (2016): 22-36.
[f] "A new and effective method for human retina optic disc segmentation with fuzzy clustering method based on active contour model". Medical & biological engineering & computing, 58(1), (2020): 25-37.
The above segmentation methods offer some alternatives to machine learning based segmentation.
2) In the related work section, a more rigorous investigation on the existing methods, such as comparison of previous approaches in terms of pros and cons, should be given. A summary table can be used in this regard.
3) Please give a frank account of the strengths and weaknesses of the proposed research method. This should include theoretical comparison to other approaches in the field.
4) The Authors need to present and discuss several solid future research directions.
Author Response
We thank the reviewer for the relevant comments on the manuscript and we have edited it to address his concerns. The changes made are detailed bellow, answering point by point to the reviewer comments, and highlighted in the modified manuscript.
- The supplementary references provided had been added to the related work (Section 3.Related Works).
- A table has been added in section 3.Related Works for the comparison with strength and weaknesses for the related works.
- The comparison has been more detailed with the comparison table.
- Future works in conclusion have been modified (Section 6.Conclusion and future works).
Please see the attachment with the updated manuscript, with highlighted changes.

Reviewer 2 Report
This paper presents a W-net for the segmentation of ARMD lesions based on well-known U-net. Even though the contribution is minor, the description of W-net is sound with reasonable experimental results. However, there are some modification necessary before next step as follows.
- In Section 4, the main description of W-Net is so brief compared to that of the other methods in Section 4.2. It is required to add more details (e.g., equations for the loss functions) in W-Net such as loss functions, etc.
- When I look through Sec. 5, it seems that training is done based on ground truth. I cannot understand that the proposal belongs to unsupervised approach. Reconstruction loss function is used often for supervised learning, so the authors need to justify that the porposal is unsupervised one.
- Since the W-Net is based on U-net, the results of U-Net should be presented in the experimental results also.
- When I check the results of Figs 5 through 10, the visual results look not so good. But the performance of F1/Precision/Recall is more than 90% for pixel-wise segmentation, which I doubt. The authors should verify those results one more time. It is helpful to add ‘Patient Id’ in Table 3 for the results of Figures 5 through 10.
- In line 253, page 7, the authors mention ‘the average dice score’ which is not in Table 1. I think dice score is useful metric to show performance comparison.
- In Table 2 & Figure 4, there is some mis-matches between ‘Gabor+KMeans’ and ‘Gabor filter +Fuzzy c-mean’.
Author Response
We thank the reviewer for the relevant comments on the manuscript and we have edited it to address his concerns. The changes made are detailed bellow, answering point by point to the reviewer comments, and highlighted in the modified manuscript.
- We added the loss function of the MSE and the soft-N-cut loss in section 4.1.
- As explained in the last paragraph of section 2.Materials, we only use the ground truth to evaluate the different models, for the only reason that even those expert annotation are not 100% reliable. The training process of the W-net has been better explained in section 4.1 as the soft-N-cut loss which make the training fully unsupervised (and therefore, only the input image is used during the training).
- The U-net training requires the ground truths, and we only compared unsupervised methods in this paper.
- Patient id has been added in Figures 5 to 10. There was effectively a mistake in the scores, especially for the Gabor Filter + Kmeans model. Results have been updated. However, the visual example for each series may not represent the performance on the entire serie, due to the growth of the GA. For instance, this is the case of the serie with Patient Id 020 : Figure 1.e) shows two images from this same serie, but with 6 years interval.
- F1 score and Dice score are the same, I should have used only one appellation.
Please see the attachment with the updated manuscript, with highlighted changes.

Reviewer 3 Report
The proposed method in this manuscript is based on the Unsupervised approach for the segmentation of dry ARMD lesions in eye fundus cSLO images. Overall this article is well written but can be improved after addressing the following comments.
1) In the abstract, the name of the dataset on which results are obtained is not mentioned. Instead of mentioning a number of images just mention the name of the dataset that will be more suitable in this section.
2) I assume that dataset collection is one of the major contributions on your part. I believe it should be mentioned in the contributions section. It would be better if you are making your dataset public for the research community.
3) In the material section, give more details about the dataset that you used in your work. Like images are collected from how many patients their average age, gender, nationalities, etc. Also, provide the link to your dataset for evaluation and comparison purposes.
4) In related work, separate heading/sub-headings should be given for the conventional methods and deep learning-based methods.
5) After related work, please include the comparison table of all the previous methods used for this problem along with their strengths and weaknesses.
6) What are the results if the supervised learning approach is used along with the simple augmentation methods for increasing the number of images and their ground truths.
Author Response
We thank the reviewer for the relevant comments on the manuscript and we have edited it to address his concerns. The changes made are detailed bellow, answering point by point to the reviewer comments, and highlighted in the modified manuscript.
- The dataset we are using is a non-public sensitive medical data provided by the XV-XX Hospital. We have the patient agreement to use them but not to publish them. Therefore, this dataset does not have a name.
- See the previous point.
- We added the number of patient this dataset is representing in section 2.Materials, but as the data have been anonymized, we do not own any private information on the patients.
- A table has been added in in section 3.Related Works, where methods has been separated between conventional methods and deep learning-based methods.
- A table has been added in in section 3.Related Works for the comparison with strength and weaknesses.
- We did not investigate the supervised learning approach because of the reliability issue of the ground truth we own. Even if it is produced by expert, due to the complexity of the task (image quality and GA complexity), the annotations obtained are not 100% reliable.
Please see the attachment with the updated manuscript, with highlighted changes.

Round 2
Reviewer 1 Report
The Authors have addressed all the comments.
Author Response
Thank you for the review of this paper.
Reviewer 2 Report
The authors revise the manuscript properly according to the comments.
I have no further comments.
Author Response
Thank you for the review of this paper.
Reviewer 3 Report
Thank you for your detailed response to my previous comments.
Here I want to include one more comment, please also include the pros and cons of your "proposed method" in the comparison table.
Author Response
Please see the attachment, the manuscript has been modified : proposed method added in the comparison table.
